
# Seismic gaps and intraplate seismicity around Rodrigues Ridge (Indian Ocean) from time-domain array analysis

Manvendra Singh[1,2] and Georg Rümpker[2]

[1] Mauritius Oceanography Institute, Avenue des Anchois, Morcellement de Chazal, Albion, Mauritius
[2] Institute of Geosciences, Goethe-University Frankfurt, Frankfurt am Main, Germany

*Correspondence to*: Manvendra Singh (msingh@moi.intnet.mu)

**Abstract.** Rodrigues Ridge connects the Reunion hotspot track with the Central Indian Ridge (CIR) and has been suggested to represent the surface expression of a sub-lithospheric flow channel. From global earthquake catalogues, the seismicity in the region has been associated mainly with events related to the fracture zones at the CIR. However, some segments of the
CIR appear void of seismic events. Here, we report on the seismicity recorded at a temporary array of ten seismic stations operating on Rodrigues Island from September 2014 to June 2016. The array analysis was performed in the time domain – by time shifting and stacking of the complete waveforms. Event distances were estimated based on a 1-D velocity model and the travel-time differences between S- and P-waves arrivals. We detected and located 63 new events, which were not reported by the global networks. Most of the events (51) are located off the CIR and can be classified as intraplate earthquakes. Local
magnitudes varied between 1.3 and 3.5. Four seismic clusters were observed along with a distinguishable swarm of earthquakes that occurred to the west of the spreading segment of the CIR during the period from March to April 2015. The Rodrigues Ridge appeared aseismic during the period of operation. The lack of seismic activity along both Rodrigues Ridge and the sections of the CIR to the east of Rodrigues may be explained by partially-molten upper-mantle material, possibly in relation to the proposed material flow between the Reunion plume and the CIR.

**1 Introduction**

The theory of plate tectonics is largely supported by the occurrence of seismicity along (oceanic) ridge systems, subduction zones and mountain belts. However, a substantial part of seismicity occurs off the ridge axes within the oceanic basins. Such events are usually denoted intraplate earthquakes (Krishna et al., 1998; Okal, 1983). Gutenberg and Richter (1941) made the first attempt to document these intraplate events. A more systematic approach was adopted by Sykes and Sbar (1973) and later
by Bergman and Solomon (1980) where they compiled a catalog of 159 oceanic intraplate events between 1939 and 1979. Assessing the seismicity inside the oceanic environment has always been a challenge as most of the permanent stations are located on the islands where the attenuation of the seismic signal along the oceanic path further reduces the detection capability. Another factor contributing to this problem is swell-generated noise in the band of 1–5 s (Okal, 1983).

Rodrigues Island is found at the eastern extremity of the Rodrigues Ridge near 19°42′S and 63°25′E, approximately 650 km east of Mauritius. The east–west trending Rodrigues Ridge was formed between 7 Ma and 10 Ma (Dyment et al., 2007; Saddul et al., 2002). It is suggested that it represents the surface expression of the interaction of the Reunion Hotspot with the Central Indian Ridge (CIR) through a sublithospheric flow channel (Duncan and Hargraves, 1990; Dyment et al., 2007; Bredow et al., 2017) though Conrad and Behn (2010) and Becker and Faccenna (2011) suggest deeper mantle circulation as the possible
mechanism of the formation of Rodrigues Ridge. Samples collected from several sites along the ridge suggest that it is composed of basaltic rocks (Dyment et al., 2001). As Rodrigues Island is approximately 250 km from the active southern part of CIR, seismicity around it is generally characterized by events recorded along this ridge system (Figure 1). The largest magnitudes, 6.3 and 6.7, were recorded in 16 August 2010 and 26 July 2012, respectively at the CIR near Rodrigues. It was



also observed that earthquakes with magnitudes 6 or greater are concentrated along Marie-Celeste fracture zone (MCFZ) only.
Krishna et al. (1998) reported six events between October and November 1984 near the ridge, on the east of the spreading segment, between Egeria fracture zone (EFZ) and MCFZ. Similarly, Bergman et al. (1984) have reported a large number of 'off-ridge' earthquakes in the region of the Southeast Indian Ridge. Interestingly, there are two segments along the CIR, between MCFZ and EFZ, for which the global catalogues are void of any seismicity (denoted GAP 1 and GAP 2 in Figure 1).

In this study we use seismological array techniques (Harjes and Henger, 1973; Husebye and Ruud, 1989; Rost and Thomas,
2002) to characterize the seismic activity around in the region of Rodrigues Ridge and to confirm the seismic gaps and provide possible explanations for them. The data for this study was collected from temporary deployment of a seismic array on Rodrigues Island between 9/2014 and 6/2016 (Figure 2).

## 2 Methods and data analysis

### 2.1 Array configuration

In order to study the seismicity around Rodrigues Ridge, we deployed an array of 10 seismic stations on the island of Rodrigues located at about 19°42′S and 63°25′E approximately 250 km west of the CIR. Rodrigues array design is based on classical 9-element arrays that use 3 and 5 seismic sensors located along two concentric rings, respectively, with an additional sensor placed in the center. The benefit of this configuration is that, with irregular sensor spacing, it provides a relatively sharp maximum of the array response function (Haubrich, 1968). For Rodrigues array, we deployed 10 sensors in a similar
configuration with the ~1.5 km radius of the inner ring and an outer ring of radius ~2.5 km (Figure 3). The final locations of the individual stations were then chosen according to the local conditions on the island (such as accessibility by roads, etc.). Each station consisted of MARK L-4C-3D sensor and an Omnirecs CUBE datalogger recording at a sampling rate of 100 Hz.

The relatively large aperture of the array (~5 km) was chosen based on events listed in the USGS database, which are located at the CIR near Rodrigues and were also recorded at the permanent station RODM. This GEOSCOPE station was relocated
during the course of the array deployment. The dominant frequency of these events is close to about 2 Hz. However, it became clear later that the dominant frequency of most earthquakes recorded by the newly-installed array is approximately 5 Hz. Therefore, we decided to perform the array analysis in the time domain, or equivalently, for a wide frequency range, to reduce possible ambiguities resulting from sidelobes in the array response function (Figure 3), as will be discussed further below.

### 2.2 Epicentral distance and origin time

For regional earthquakes, the slowness (apparent velocity) cannot be used to determine the epicentral distance of an event, as the raypath is mainly confined within the uppermost mantle and the depth variations of velocity are not well constrained. We, therefore, use the arrival-time difference between the S and P waves in conjunction with a simplified 1-D velocity model of the crust and upper mantle to approximately determine the epicentral distance.

In this model, we keep the hypocentral depth fixed at 6 km and the raypath corresponds to a head wave (Figure 4). From
receiver-function analysis (Fontaine et al., 2015), a Moho depth of 10 km has been determined beneath Rodrigues Island. We fix the crustal thickness in our laterally-homogeneous model to this value, such that the thickened oceanic crust is accounted for on the receiver side leg of the raypath. For the P-wave velocity in the crust, we assume $V_P^C = 6.1\,\mathrm{km/s}$ and in the mantle $V_P^M = 7.9\,\mathrm{km/s}$ with a $V_P/V_S$ ratio of 1.80 based on the values as suggested by Christensen (2004); Kong et al. (1992); Wolfe et al. (1995) and Grevemeyer et al. (2013). We explored the influence of these parameters on the determination of the epicentral
distances for the events that occurs at the distances of ~120 and ~265 km from the array. As shown in Figure S1 of supporting

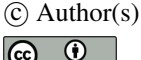



information, results are most sensitive to variations of the P-wave velocity in the mantle and $V_P/V_S$, owing to its contribution to the raypath.

The final value for the epicentral distance of an event (with respect to the center of the array) is obtained by taking the mean of the distance values determined at all stations of the array. Stations for which the recordings do not exhibit a clear onset for

either the P- or S-phase are discarded from this calculation. The standard deviation (SD) is used to definethe error of the distance calculation.

To determine the event origin time, $t_{origin}$, the travel time, $t_{tt}$, was calculated from the distance obtained from the S- and P-wave travel time difference for the central (reference) station. By subtracting this value from the manually-picked P-wave arrival time, $t_P$, we obtain $t_{origin} = t_P - t_{tt}$ , where it is assumed that $t_P$ is given as an absolute time value.

**2.3 Magnitude determination**

In order to calculate the magnitude of an event, the instrument response is removed from the velocity seismogram, which is then convolved with the Wood–Anderson transfer function to obtain the ground motion in nanometers. Magnitudes are determined using all available stations for which the recordings show a clear (dominant) S-phase after convolution with Wood–Anderson transfer function. Again, the mean and the SD of the

magnitude value is calculated. As most of the events are located well within 1000 km radius, we use the relation given by Havskov and Ottemoller (1999) in SEISAN package to determine the local magnitude

$$M_L = \log_{10}(A) + 1.1 \times \log_{10}(\Delta) + 0.00189 \times \Delta - 2.09 \tag{1}$$

where $A$ is the amplitude (in nm) and $\Delta$ is the epicentral distance (in km).

**2.4 Beamforming and array transfer function**

In seismic array analysis, the slowness and the backazimuth of an event are determined by beamforming. Assuming a plane wave front of horizontal slowness $s_0$ moving across the array with an apparent velocity ($v_a = 1/|s_0|$), the waveform at station $j$ is given by

$$w_j(t) = w(t - r_j \cdot s_0) \tag{2}$$

where $r_j$ defines the position of the station with respect to a suitable coordinate system.

For an array consisting of $M$ stations, the beam energy is calculated from the trace amplitudes within a suitable time window defined by $t_1$ and $t_2$ according to (e.g., Harjes and Henger, 1973; Rost and Thomas, 2002)

$$E = \int_{t_1}^{t_2} y^2(t)\, dt = \int_{t_1}^{t_2} \left[ \frac{1}{M} \sum_{j=1}^{M} w_j(t + r_j \cdot s) \right]^2 dt \,, \tag{3}$$


where $s$ denotes the (trial) slowness for the current beam. The beam energy reaches a maximum, if $s = s_0$ . The backazimuth is then obtained from slowness components according to $\tan^{-1}(s_{0x}/s_{0y})$ where $x$ and $y$ correspond to the eastern and northern components, respectively (see details given below).

In view of equation (1) and using Parseval's theorem in combination with the shift theorem of Fourier transforms, we can write the energy in the form



$$E = \frac{1}{2\pi} \int\limits_{\omega_1}^{\omega_2} \overline{y}^2(\omega)\, \mathrm{d}\omega = \frac{1}{2\pi} \int\limits_{\omega_1}^{\omega_2} \overline{w}^2(\omega)\, C(\omega, s-s_0)\, \mathrm{d}\omega \tag{4}$$

where the bar denotes Fourier-transformed function and the array transfer function, $C(\omega, s-s_0)$, in the frequency-slowness

domain is given by (e.g. Schweitzer, 2002)

$$C(\omega, s-s_0) = \left[ \frac{1}{M} \sum_{j=1}^{M} e^{i\omega r_j \cdot (s-s_0)} \right]^2. \tag{5}$$

It is further assumed that $\overline{w} = 0$ is outside of the range of integration used in equation (4). The array transfer function defines
the sensitivity and resolution of the array for seismic signals with frequency $\omega$ (Figure 3).

**2.5 Data example**

All the events for this study were detected by manual inspection of hourly traces from all stations of Rodrigues array using the
SEISAN package (Havskov and Ottemöller, 1999). A time window of 120 s was cut around each event using the GIPPtool

software (http://www.gfz-potsdam.de).

As shown in Figure 5, we estimate the epicentral distance from the arrival-time differences of the S and P waves. The STA/LTA
ratio of the Z-component trace is calculated to aid in the manual picking of the two arrivals. The mean of the P-phase picking
time, calculated from all the picks, is used to determine the time window to calculate the beam energy later in the array analysis.
For this example, the arrival-time difference is 13.3 s, which corresponds to a distance of approximately 119 km (Table S1).

Finally, the magnitude of the event is calculated from equation (1). Here, we obtain a magnitude 3.0. In the following, we
apply this methodology to all events detected that exhibit a clear P and S-wave onset.

Conventionally, array analyses are performed in the frequency domain, which is computationally advantageous as the energy
stacking can be limited to the dominant frequency or a narrow frequency band. As explained above, however, we perform the
array analysis in the time domain to include the complete waveform of the first arrival. The time-domain analysis corresponds

to a broad-band energy stack and suppresses the effects of unwanted sidelobes, but there is an additional benefit: the frequency
domain approach usually requires selection of a common time window for all traces before the Fourier transform is applied.
In cases of significantly different arrival times of the phase to be analyzed (e.g., due to a large aperture), a relatively wide
common time window has to be selected such that the cut waveforms of individual traces may be significantly different. In the
time domain, however, we can time shift the traces (with respect to the trial slowness value) before stacking, which may then

be performed within a much narrower time window. This approach ensures that only the relevant waveform is contained within
the stack, provided that the correct time shift has been applied.

Our analysis involves applying time shifts to all the traces with respect to reference trace (no. 1) for different values of
slowness, defined by a grid, and calculating the energy of the resulting stacked trace or beam within the time window of the
first arrival which is selected using mean of the P-wave picks on all the usable traces for a given earthquake. Only vertical (Z)

component traces from all stations were used for the beamforming.

Figure 6(a) shows the Z-component traces for an earthquake on 6 April 2015 originating at 20:25:42h. A band pass between 2
and 10 Hz is applied. The trace number on the vertical axis corresponds to the respective station number. The amplitudes of
trace 3 are set to zero as it did not show a clear event.

At first, a relatively wide time (~2 s) window is automatically chosen to zoom-in on the P-phase (Figure 6b). Then, a narrower
time window (~ 0.6 s or about three dominant periods wide, as indicated by the red vertical lines) is automatically selected





with respect to reference trace 1 based on the P-phase picking on all the traces. Time shifts are applied to all other (normalized) traces in correspondence to the slowness values of the grid which is used in the energy stack (Figure 6c). The stack, however, is only calculated from the trace segments within the narrow time window.

A predefined grid with slowness from −0.3 to 0.3 s/km, equally spaced over 248 × 248=61,504 points is used for calculating the appropriate time shifts for each trace and the resulting energy. The red circle in Figure 6c denotes the maximum beam energy at $s_0 = (0.06, 0.07)$ which corresponds to a backazimuth $\Phi \sim 38°$ and $v_a \sim 10.6\,\text{km/s}$. We assume that this relatively large value for the apparent velocity results from steepening of the raypath at shallow depth beneath Rodrigues Island (where the propagation velocity is smaller than 6.1 km). The corresponding shifted traces and the beam (by summing the amplitudes
of all traces) are shown in Figure 6d.

The values of slowness components thus obtained, were used to calculate the corresponding apparent velocity, $v_a$, and the backazimuth, $\Phi$ of the event according to

$$v_a = 1/\sqrt{s_{0x}^2 + s_{0y}^2} \quad \text{and} \quad \varphi = (180°/\pi)\tan^{-1}(s_{0x}/s_{0y}) \tag{6}$$


where $s_{0x}$ and $s_{0y}$ are the values of slowness vector obtained for the beam with maximum energy.

To estimate the error in backazimuth, $\delta\Phi$, we define the area enclosed by the contour at a level of 95% of the maximum beam energy as a confidence region. The error in backazimuth in terms of kilometers ($\delta\Phi_{km}$) is then given by

$$\delta\varphi_{km} = (\pi/180°)\,\delta\varphi\,\Delta \tag{7}$$

where $\Delta$ is the epicentral distance (in km) and $\delta\varphi$ is given in degrees.

## 3 Comparison with event locations from global networks

The larger earthquakes along the CIR picked up by the global networks are listed in the earthquake database provided by USGS (https://earthquake.usgs.gov/earthquakes/search/). Thirty events reported in the catalog were also recorded by Rodrigues array. Six of these events exhibit sufficient signal-to-noise ratio to perform the array analysis for a comparison with
the USGS-reported locations.

The mean distance of the event was calculated using distances from all the stations that provided clear P- and S-phases. Figure 7 shows the 3-component seismogram and the STA/LTA trigger function to identify the onset times. The arrival-time difference of ~24.63 s corresponds to a distance of ~231 km (based on the model described above). For the magnitude, we derive $M_L = 3.1$.
Geographical coordinates, origin times, and magnitudes for the six events are shown in Table 1. They can be compared with the results provided by the USGS database (Table 2). Magnitudes obtained from the array analysis are slightly lower than those reported by USGS. This may be in part due to different magnitude scales used (local magnitude versus body wave magnitude). Ristau (2009) compared different magnitude scales for New Zealand earthquakes, where $m_b$ was lower than $M_L$ for deep focus (>33 km) events but the results were fairly consistent for shallow (≤33 km) earthquakes. In the current study,
we consider that the data is not sufficient to derive a relationship between $m_b$ and $M_L$.

The array analysis for the six events was performed as described above. An example is given in Figure 7 for an event of 14 February 2015 at 07:08:59h. P and S phases are clearly visible and the dominant frequency of the P-wave arrival for this event is about 3.5 Hz. From the array analysis, we obtain a clear maximum for $s_0 = (0.07, 0.08)$ which corresponds to a backazimuth $\Phi \sim 39°$ and apparent velocity $v_a \sim 9.5\,\text{km/s}$. The time shifts derived from the obtained slowness value according to $\delta t_j = \delta r_j \cdot s_0$



(where $\delta r_j$ corresponds the position of station j with respect to the reference station) yield a good alignment of the P-wave signal.

For three events, we obtain small variations in backazimuth, which are well within the error estimations. Events on 24 November 2014 at 22:23:23, 2 April 2016 at 18:01:49h and 2 June 2016 at 21:18:10h show differences of about 11°, 9° and 7°, respectively. Array-derived distances from the reference station are generally smaller than those given in USGS catalog

except for the event of 02 April 2016 at 17:53:21h, where the distance obtained from array analysis is well within the error range.

We attribute these differences to local inhomogeneities not accounted for in the array analysis and to the simple 1-D model used for the distance estimates (in addition to possible errors in the global locations). Figure 8 shows a comparison of the results obtained by the array analysis with those provides by the USGS catalogue. Generally, the results agree well. On average,

location differences are about 17 km, which is a reasonable value considering the uncertainties of the approach.

## 4 Results and Discussion

Using the array technique, we were able to detect and locate 63 earthquakes in the Rodrigues–CIR region that are not reported by any global network (Figure 9). The details of all the events, such as event location, origin time, and magnitude are summarized in Table S1 (supporting information). The magnitudes of these events range from 1.3 to 3.5 and are spread out in

a region of radius up to 600 km from the array. The nearest event to Rodrigues Island ($M_L$ 1.5) occurred on 12 February 2016 at 19:21:45h, to the north of the island at a distance of about 36 km.

Of the 63 events, 51 are located off the ridge axis and can therefore be classified as intraplate earthquakes. Twelve events were located very close to the ridge axis and are not being considered as intraplate events. Twenty-four events occur between backazimuths of 29° and 42° at a distance of about 120 km and exhibit magnitudes between 1.3 and 3.0 (Figure 9). Almost all

events in this backazimuths (20) occur from March to April 2015 (Figure 10). Magnitudes for this cluster of earthquakes are variable and do not follow a certain (main shock–after shock) pattern (Cluster 1, Table S2 of supporting information). Detailed bathymetry indicates a step-like non-transform discontinuity at the south-eastern end of this chain of events.

Three more clusters occur to the east, south and to the north-west of Rodrigues at distances of about, 265, 140 and 220 km with 6, 3 and 7 events, respectively. This is further corroborated by Figure 10, where events' longitude and latitude are plotted

as functions of time. Also shown is the monthly number of events, which exhibits a clear maximum for March 2015 in relation to the activity of Cluster 1. As Cluster 4 (220 km east of Rodrigues Island) is close to the ridge axis, events of this cluster are not considered as intraplate events.

As shown in the sensitivity test provided in the supplementary information (Figure S1), the location errors for Cluster 1 are much smaller as compared to Cluster 3, partially owing to their distance from the Rodrigues array. Some influence of

anisotropic velocity variation may also be possible, as studies by Barruol and Fontaine (2013) and Scholz et al. (2018) suggest fast-axis direction trending east–west around the Rodrigues–CIR region.

Various mechanism providing explanation to the cause of intraplate seismicity have been proposed previously. De Long et al. (1977) suggested that due different ages across the fracture zones, as observed in the Rodrigues–CIR region, may have differential subsidence causing stresses and hence earthquakes. Similarly, Collette (1974) and Turcotte (1974) suggested

thermal contraction of the oceanic crust as another possible mechanism. Mantle-derived carbon dioxide discharge (e.g., Bräuer et al., 2003; Gold and Soter, 1984/85; Irwin and Barnes, 1980) can also provide an explanation for the intraplate seismic activity around the Rodrigues–CIR region as most of the detected events occur in clusters. In the continental region, similar clustering has been associated with carbon-dioxide discharge (e.g., Lindenfeld et al., 2012).

It is interesting to note that of the two seismic gaps along the CIR, as indicated above, still show no seismic activity based on

the new data. Thus, these seismic gaps, may indeed, represent an anomalous section of the CIR that is deforming aseismically.





The anomalous nature of this region is further corroborated by the fact that globally-recorded events from the further southern section of CIR are not detected by the array (see events south of -20° in Figure 8). This could be explained by a more extended region of partially-molten material in the upper mantle that causes significant attenuation of wave amplitudes for the corresponding raypaths (Figure 11), as also suggested by Mazzullo et al. (2017). In combination with the absence of any

seismic activity along Rodrigues Ridge, this may be taken as evidence for the explanation of Rodrigues Ridge as a surface expression of the interaction of the Reunion Hotspot with the CIR through a sublithospheric flow channel (Morgan, 1978; Dyment et al., 2007; Bredow et al., 2017). Another explanation for the lack of seismicity in GAPs 1 and 2 could be excessive magmatism and a resulting     relatively thin lithosphere which does not support large earthquakes (see Cannat, 1996; Grevemeyer et al., 2013). However, as we were able to detect relatively small events with Rodrigues a thinner lithosphere

seems a less likely explanation.

### 5 Conclusions

We installed a 10-station seismological array on Rodrigues Island to study the seismicity along a remote section of the CIR and nearby areas including Rodrigues Ridge. The results show that array analysis provides a valuable tool to study earthquake activity in oceanic regions which are relatively inaccessible otherwise. The region around Rodrigues Island clearly shows

evidence of intraplate seismicity. Of the 63 events detected by Rodrigues array, the majority are located within a cluster at a distance of ~120 km from the island with backazimuths between 29° and 42°. The local magnitude ($M_L$) of the events detected range between 1.3 and 3.5. Three additional event clusters were identified. A possible explanation for this seismic activity is $CO_2$ degassing from the mantle as most of the events are distributed within these clusters instead of being distributed more linearly along fault zones. The lack of seismic activity along both Rodrigues Ridge and a section of the CIR to the east of

Rodrigues (GAP 2) may be explained by partially-molten upper-mantle material, possibly in relation to the proposed material flow from the Reunion plume and the CIR (Morgan, 1978; Dyment et al., 2007; Bredow et al., 2017). This explanation is further supported by the observation that relatively strong seismic events from the CIR, east to south–east of Rodrigues (which appear in the USGS catalogue) are not detected by the array. However, a detailed geodynamic model for the ridge–plume interaction is still needed. We anticipate that longer-term deployments of seismic arrays on Rodrigues and other remote islands

of Mauritius, such as Agalega and St. Brandon, will provide further constraints on the seismic gaps along the CIR and the intraplate seismicity of the region. Dedicated deployments of Ocean Bottom Seismometers (OBS) at or near these targets are another option for future studies.

### Acknowledgments

The study presented here was funded by the Mauritius Oceanography Institute, Mauritius, through Ministry of Finance and by

a DFG grant to G.R. The instruments for the temporary networks were provided by the Geophysical Instrument Pool Potsdam at Deutsches GeoForschungsZentrum, Potsdam. The data will be publicly available from GEOFON data archive of GeoForschungsZentrum as from July 2021. We wish to thank the Government of Mauritius for allowing us to carry out the study, as well as the MOI and Rodrigues Regional Assembly (RRA) for the support provided during the fieldwork. We thank J. Dyment of IPGP for his suggestions on the origin of seismicity in Cluster 1. We also thank Frederik Link, Corrado

Surmanowicz, Olivier Pasnin and Shane Sunassee for providing help during fieldworks in Rodrigues.

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





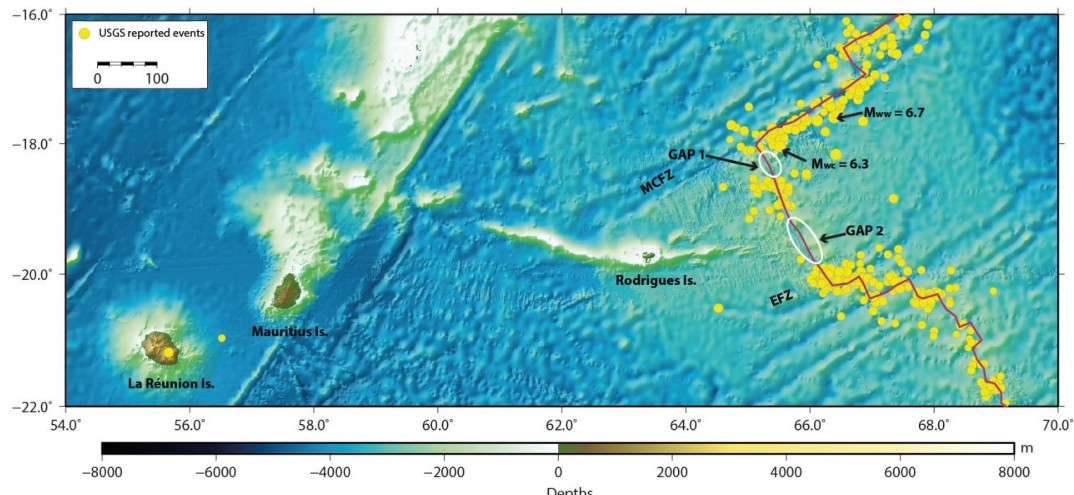

**Figure 1:** Tectonic setting of Rodrigues–Central Indian Ridge (CIR) region. The ridge axis is shown by solid line (red). Event locations from USGS catalog (2000/01–2019/02) are shown in yellow. MCFZ, Marie-Celeste fracture zone and EFZ, Egeria fracture zone. The gaps in seismicity (GAP 1 and GAP 2) along the section of CIR between MCFZ and EFZ have been marked by white ellipses.


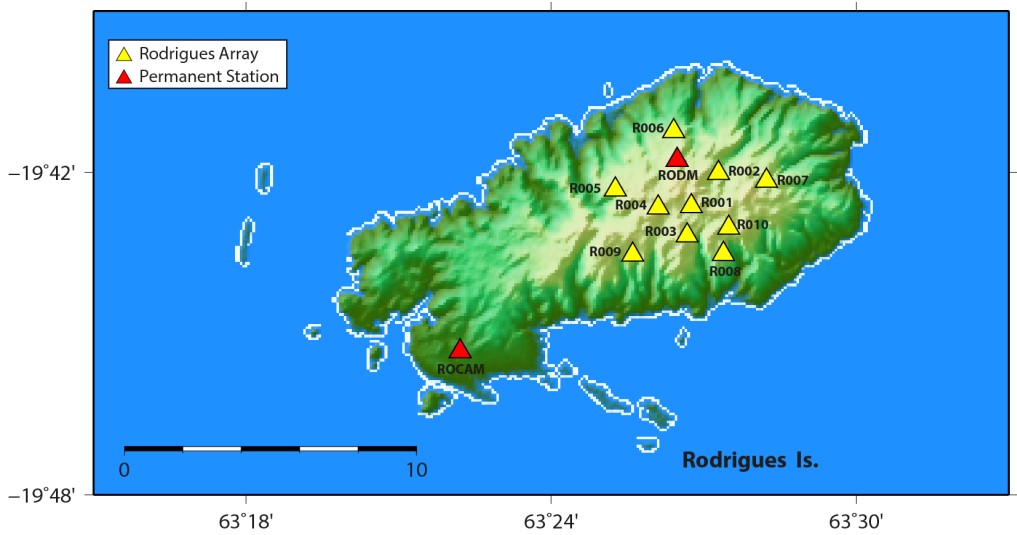

**Figure 2:** Locations of individual stations in Rodrigues array (yellow triangles). Red triangles denote permanent stations ROCAM and 350 RODM. Station RODM was operational between 10/11/2010 and 07/09/2014.





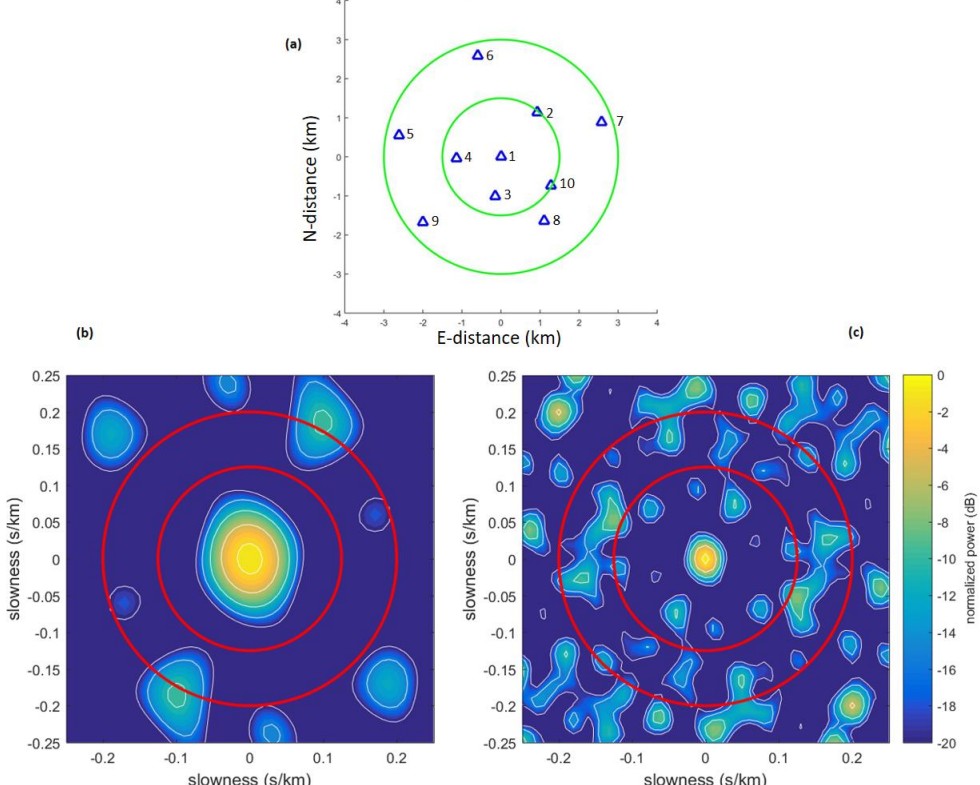

**Figure 3:** Configuration of the seismic array deployed on Rodrigues Island (a). Station locations are indicated by blue triangles. The green circles represent inner and outer rings of the array with a radius of ~1.5 and ~2.5 km, respectively, from the central (reference) station. The array transfer function of the Rodrigues array is shown at 2 Hz (b), and 5 Hz (c). The inner and the outer red rings correspond to an apparent velocity of 8 and 5 km/s, respectively. In the real-data analysis, the apparent velocity at which the maximum occurs, is used as a first indication to discriminate between crustal and upper-mantle raypaths.

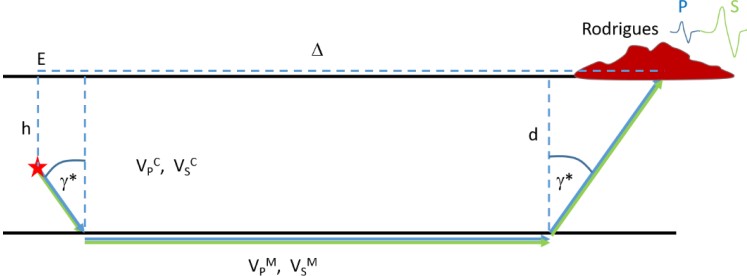

**Figure 4:** Cartoon depicting the model employed to estimate epicentral distances from travel-time differences between S- and P-wave arrival. The simple 1-D velocity model of the oceanic crust and uppermost mantle consists of a uniform layer overlying a uniform half-space. The parameters are set as follows: thickness of the crust, $d = 10$ km; hypocentral depth, $h = 6$ km; P-wave velocity in the crust, $V_P^C = 6.1$ km/s ; P-wave velocity in the mantle, $V_P^M = 7.9$ km/s ; $V_P/V_S = 1.80$ in both crust and mantle. Figure S1 (supporting information) shows the influence of these parameters on the determination of the epicentral distance.



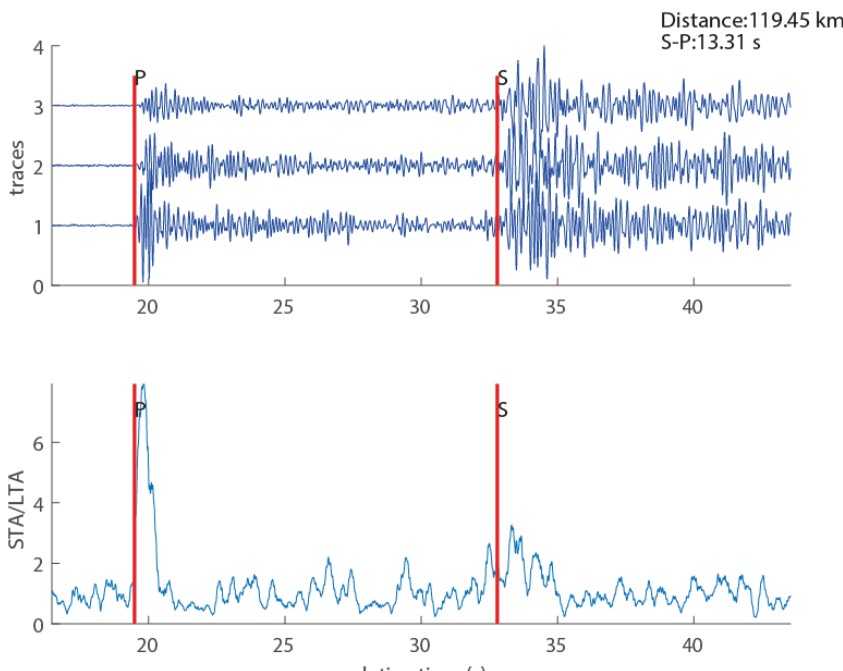


**Figure 5:** Estimating the epicentral distance from the arrival time difference between S- and P waves at the for reference station 1. The vertical, North and East components are shown by traces 1, 2 and 3, respectively (upper panel). The vertical red lines panel show manually-picked P- and S-phases. The picking is aided by calculating the STA/LTA of the Z-component (lower panel).





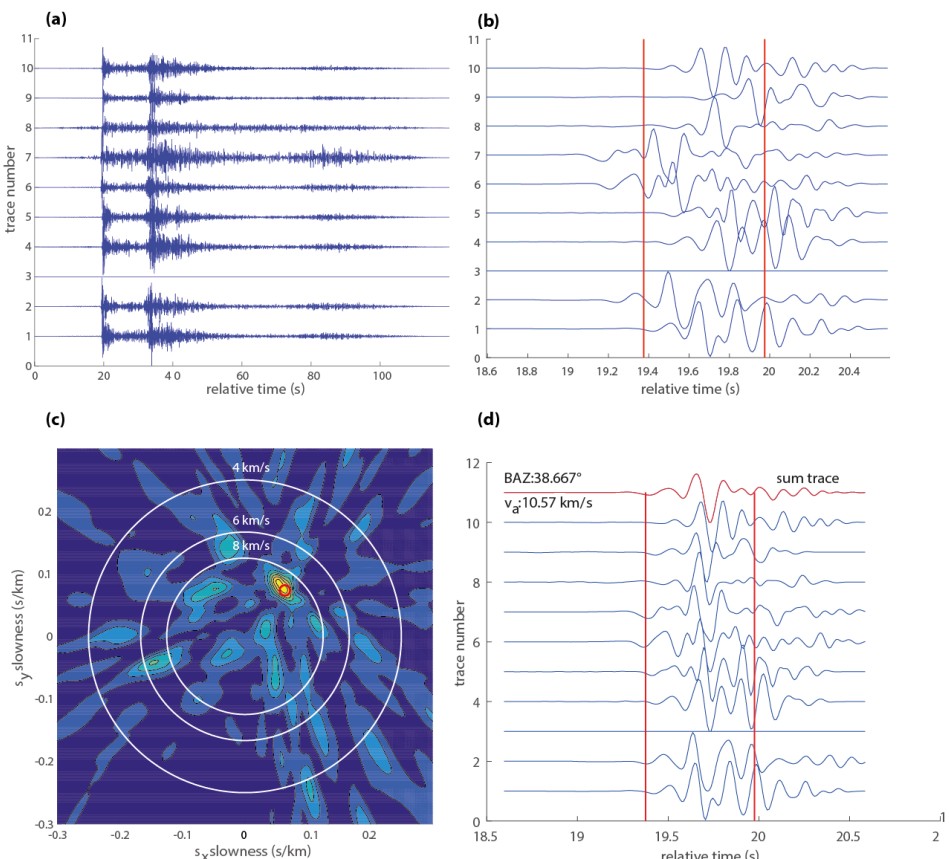

**Figure 6**: Array analysis of a regional event near Rodrigues. (a) Vertical-component traces; station 3 was not recording at the time of the event. (b) Zoom-in of the P-waveforms. After applying the time shift corresponding to a trial slowness $s$, the beam energy is calculated from squared amplitudes within the time window enclosed by the red lines. (c) Beam energy as function of horizontal slowness components $s_x$ and $s_y$. The maximum is marked by a red line and corresponds to the slowness $s_0$. White circles denote apparent velocities, $v_a$, of 4, 6, and 8 km/s. (d) Shifted traces and beam (red trace) corresponding to the maximum indicated in (c). The backazimuth of this event is determined at about 38° with an apparent velocity of 10.57 km/s.





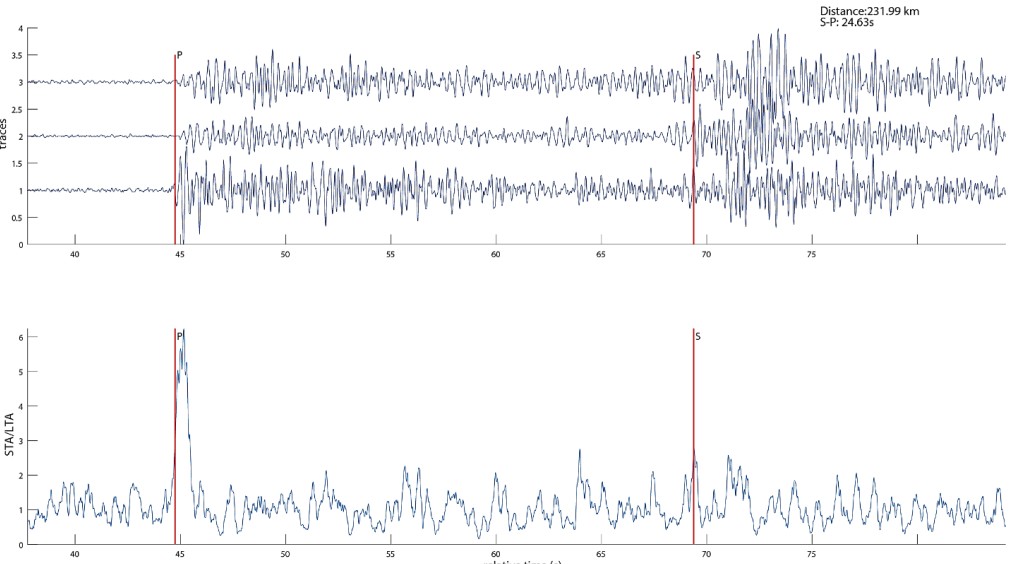

**Figure 7:** Estimating the epicentral distance from the arrival time difference between S- and P waves at the reference station 1. The vertical, North and East components are shown by traces 1, 2 and 3, respectively (upper panel). The vertical red lines panel show manually-picked P- and S-phases. The picking is aided by calculating the STA/LTA of the Z-component (lower panel).






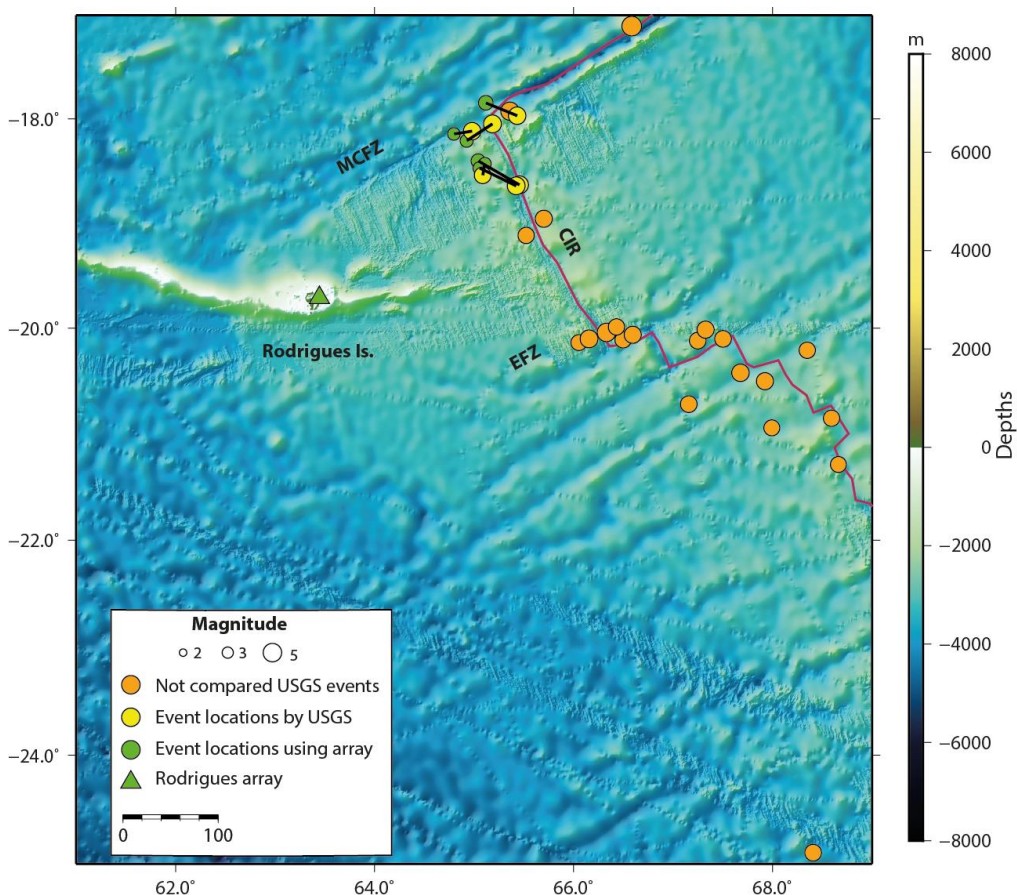

**Figure 8:** Event locations from the array analysis in comparison to those from the USGS database between September 2014 and June 2016. The black bars pair the respective events. Average and maximum separations between the locations for the same event are ~17 and ~32 km, respectively. For most of the events reported in the catalogue, no reliable location could be obtained using Rodrigues array (see section 3). MCFZ: Marie-Celeste fracture zone; EFZ: Egeria fracture zone.



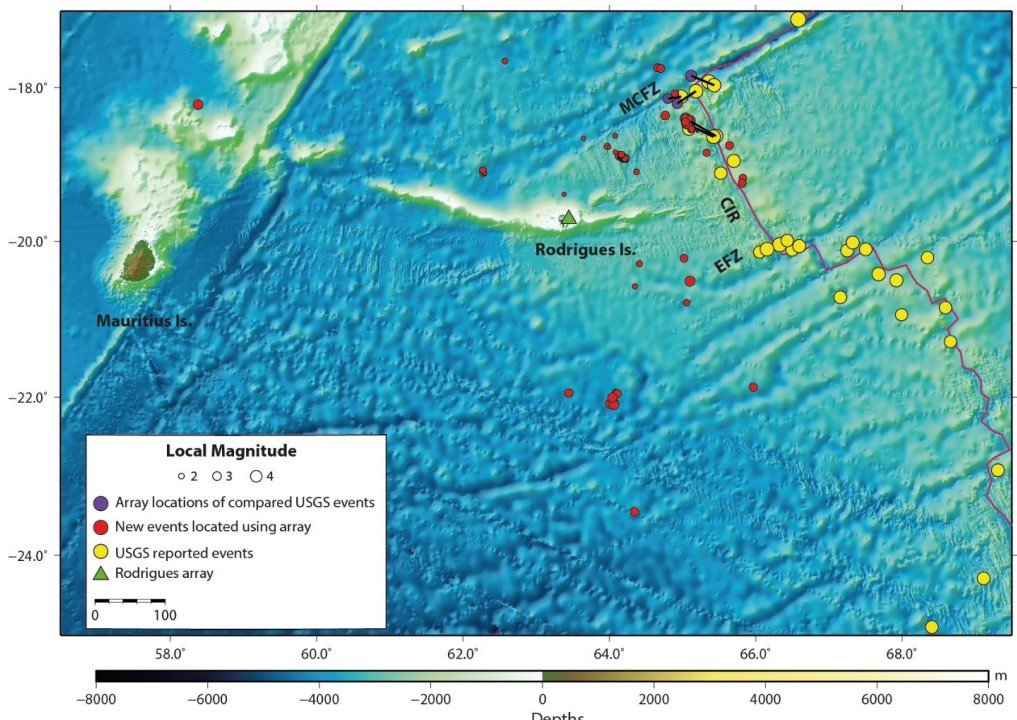

**Figure 9**: Locations of new events detected and located using array methods are shown by red circles whereas yellow circles denote events from USGS catalog for the same duration as the deployment of Rodrigues array. Purple circle represents event locations from the array analysis in comparison to those from the USGS database between September 2014 and June 2016. The black bars pair the respective events.
CIR, Central Indian Ridge; MCFZ, Marie-Celeste fracture zone and EFZ, Egeria fracture zone.



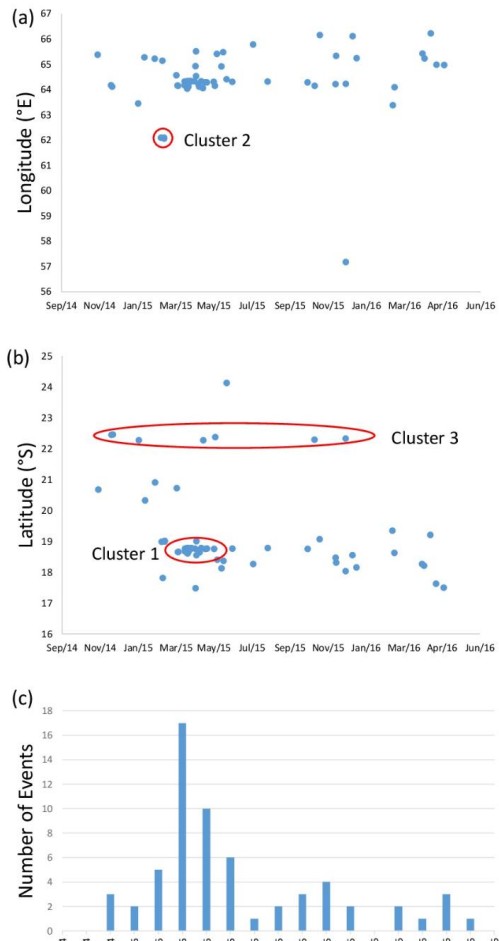

**Figure 10:** Variation in longitude (a) and latitude (b) of the events detected and located using Rodrigues array. Monthly distribution of the
events is shown in (c). The red solid line in (b) marks the events for Cluster 1 (~120 km north-east of Rodrigues Island), Cluster 2 (~1 km
north-west of Rodrigues Island), Cluster 3 (~265 km south of Rodrigues Island), and Cluster 4 (~220 km north-east of Rodrigues Island).



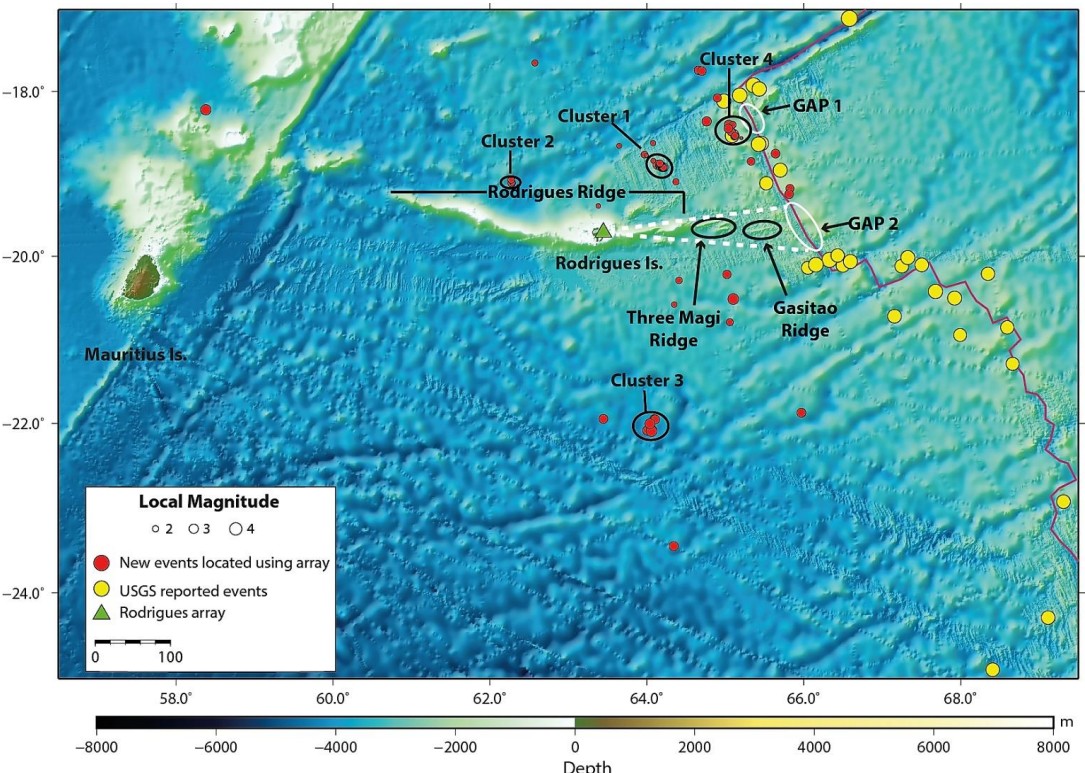

**Figure 11:** Tectonic setting of Rodrigues–Central Indian Ridge (CIR) region. The ridge axis is shown by solid line (red). Event locations from USGS catalog (2000/01–2018/01) are shown by yellow circles whereas events located using Rodrigues array (2014/09–2016/06) are shown by red circles. The empty GAPs 1 and 2 have been demarcated by white ellipses. The white dash line marks the possible region of partial molten material below the eastern extension of Rodrigues Ridge.

**Table 1: Events from USGS catalogue located using Rodrigues' array**

| Date,time (yymmdd,hh:mm:ss) | Lat. (°) | Lon. (°) | $M_L$±SD | Dist.±SD (km) | BAZ (°) | BAZ error Min (°) | Max (°) |
|---|---|---|---|---|---|---|---|
| 141124,22:23:23 | −18.404 | 65.043 | 3.7±0.1 | 222.02±3.23 | 49.39 | 46.73 | 53.87 |
| 150214,07:08:59 | −18.100 | 64.837 | 3.1±0.3 | 231.27±5.01 | 39.48 | 35.55 | 44.06 |
| 150214,11:13:33 | −18.210 | 64.927 | 3.2±0.1 | 228.26±2.22 | 43.27 | 38.67 | 45.00 |
| 160402,17:53:21 | −18.432 | 65.108 | 3.1±0.4 | 225.21±5.66 | 51.14 | 46.73 | 54.60 |
| 160402,18:01:49 | −18.472 | 65.057 | 3.3±0.1 | 218.24±3.89 | 51.14 | 47.64 | 55.61 |
| 160602,21:18:10 | −17.846 | 65.118 | 3.5±0.0 | 271.93±5.75 | 40.61 | 36.13 | 43.27 |



**Table 2: Details of events from USGS catalogue for comparison**

| Date,time (yymmdd,hh:mm:ss) | Lat. (°) | Lon. (°) | $m_b$ | Dist. (km) | BAZ (°) |
|---|---|---|---|---|---|
| 141124,22:23:19 | −18.6335 | 65.4567 | 4.8 | 242.74 | 60.78 |
| 150214,07:08:55 | −18.1178 | 64.9817 | 4.7 | 239.68 | 42.62 |
| 150214,11:13:29 | −18.0478 | 65.1849 | 4.8 | 260.07 | 44.99 |
| 160402,17:53:18 | −18.5407 | 65.0872 | 4.5 | 215.96 | 53.24 |
| 160402,18:01:45 | −18.6435 | 65.4247 | 4.8 | 239.27 | 60.61 |
| 160602,21:18:07 | −17.9694 | 65.4337 | 4.7 | 284.98 | 47.54 |