# Peer review of "Seismic gaps and intraplate seismicity around Rodrigues Ridge (Indian Ocean) from time-domain array analysis"

_Solid Earth, 2020_

## Referee Comment (RC1) · Anonymous Referee #1 · 22 Jun 2020

General comments

The manuscript presents a thorough array-based analysis of seismicity surrounding Rodrigues Island in the Indian Ocean. Due to its remote location and proximity to the Rodrigues Triple junction, this data set offers interesting insights into intra-plate seismicity of an oceanic plate. The authors exploit the array geometry to estimate locations of 63 events that were not detected by global networks. They suggest a simple, but adequate way of estimating epicentral distance and use backazimuths derived from array analysis. The method is very clearly documented and well-illustrated. The results suggest some well-defined clusters of intra-plate seismicity, that are clearly located away

from any of the surrounding ridges. In addition, the authors discuss gaps in the seismicity distribution along the Central Indian Ridge, that are not closed by their additional data. While all results are clearly described, the interpretation is somewhat short and could easily be enhanced by some more information.

Specific comments

To expand the discussion of the seismicity, the authors could for example include in their interpretative Fig. 11 the age of the oceanic lithosphere to discuss whether age differences and hence differing thermal contraction could be responsible for the intra-plate seismicity. Clusters 2 and 3 seem to be located in prolongation of transform zones, so this could support this interpretation. Cluster 1, in contrast, shows no apparent relation to different lithospheric ages, nor does cluster 4, which is probably rather located on the Central Indian Ridge. An important observation, that supports the interpretation of a potential melt channel between Rodrigues Island and the seismic gap 2 on the Central Indian Ridge, is the lack of arrivals from earthquakes recorded along the Central Indian Ridge south of this gap, while other earthquakes at comparable distances have been detected on Rodrigues Island. To make this more obvious, Fig. 11 could include a colour or symbol coding to distinguish between earthquakes in the global catalogue that were detected and those that were not detected. A seismic stripe-and-gap pattern in teleseismic and hydroacoustic data have for example been discussed in more detail by Escartin et al. (2008) and by Simao et al. (2010). They see a relative lack of earthquakes near magmatic centres of spreading segments, whereas segment ends at the Mid-Atlantic Ridge tend to show increased seismicity rates. This observation could provide support for the interpretation of gap 2. A regional seismicity analysis with data just south of the survey area (25°S) is presented by Tsang-Hin-Sun et al. (2016). These authors also find seismic gaps that are even somewhat clearer delineated since they use a hydroacoustic data set with lower detection thresholds. This study could therefore provide addditional evidence for the existence of the seismic gaps in the present study area. It may also be advisable, when delineating seismic

gaps along the Central Indian Ridge, to include an as long observation span as possible from teleseismic catalogues. Fig. 11 uses 18 years of data from the USGS catalogue. The reviewed ISC bulletin shows many more events in this area for the period 1970-2017. It becomes obvious that the entire section between 18°S and 20°S is relatively aseismic compared to the ridge sections to the north and south. Gap 1 and 2 seem to be separated only by one very distinct cluster of teleseismic earthquakes that potentially coincides with cluster 4. It would be interesting to see whether the seismicity of cluster 4 is swarm-like and occurs in a short time period. Earthquake swarms may point to ongoing magmatism and further support the interpretation of a melt region. At very slow spreading rates, even strong earthquakes detected at teleseismic distances may be an indicator of magmatic activity (Müller & Jokat, 2000; Schlindwein, 2012).

References: Escartin, J., D. K. Smith, J. Cann, H. Schouten, C. H. Langmuir, and S. Escrig (2008), Central role of detachment faults in accretion of slow-spreading oceanic lithosphere, Nature, 455 , 790-794. MuÌĹller, C., and W. Jokat (2000), Seismic evidence for volcanic activity discovered in central Arctic, Eos, 81 (24), 265, 269. Schlindwein, V. (2012), Teleseismic earthquake swarms at ultraslow spreading ridges: indicator for dyke intrusions?, Geophys. J. Int. , 190 , doi: 10.1111/j.1365-246X.2012.05502.x. Simão, N., J. Escartín, J. Goslin, J. Haxel, M. Cannat, and R. Dziak (2010), Regional seismicity of the Mid-Atlantic Ridge: observations from autonomous hydrophone arrays, Geophys. J. Int., 183, doi: 10.1111/j.1365-246X.2010.04815.x. Tsang-Hin-Sun, E., J.-Y. Royer, and J. Perrot (2016), Seismicity and active accretion processes at the ultraslow-spreading Southwest and intermediate-spreading Southeast Indian ridges from hydroacoustic data, Geophys. J. Int. , 206 , doi: 10.1093/gji/ggw201

---

## Referee Comment (RC2) · Anonymous Referee #2 · 19 Jul 2020

General comments: The authors use a temporary array on Rodrigues Island to detect and locate (mostly) intraplate earthquakes west of the Central Indian Ridge. This is an entirely new study, which makes the most out of a small array of seismic stations. The analysis is sound; the approach of using beam forming to estimate azimuth and S-P times to estimate distance is sensible.

Specific comments: It is not clear that these locations provide much new information about seismic gap 2, given the absence of detection of events from the adjacent Egeria FZ. The lack of small events could be attributed instead to poor propagation of Pn and Sn along the path to the array. There is no need for the direct effects of melt on

attenuation along the paths; the attenuation is more likely attributable to the presence of thin lithosphere associated with the flow of hotter mantle to the spreading center from the hotspot.

Technical comments:

Lines 176-179 appear to be out of place, since they refer to "the event", which is not introduced until lines 186-187. Also lines 186-191 suggest that array analysis of this example event is shown in figure 7, but that aspect of the event is not illustrated.

Can comparison of USGS event locations with the array locations be used to find a consistent P time correction for each station?

Since Figure 9 largely duplicates information shown in Figures 8 and 11, this figure could be used to show regions of uncertainty around each of the locations calculated from uncertainty in azimuth and scatter in S-P picks. P picks appear to be pretty clear, but S picks somewhat subjective. Are S picks on each seismogram made independently, or is there iteration between different picks for each event?

Cluster 4 is not shown in Figure 10, although it is mentioned in the caption.

---

## Author Comment (AC1) · 14 Aug 2020

We would like to thank the reviewers for the time spent on the review and for the helpful comments and constructive suggestions. We discuss below the comments made by referee #1. The comments of the referee are in black and the replies from Authors are provided in blue to facilitate the reading. General comments Referee 1: The manuscript presents a thorough array-based analysis of seismicity surrounding Rodrigues Island in the Indian Ocean. Due to its remote location and proximity to the Rodrigues Triple junction, this data set offers interesting insights into intra-plate seismicity of an oceanic plate. The authors exploit the array geometry to estimate locations of 63 events that

were not detected by global networks. They suggest a simple, but adequate way of estimating epicentral distance and use backazimuths derived from array analysis. The method is very clearly documented and well-illustrated. The results suggest some well-defined clusters of intra-plate seismicity, that are clearly located away from any of the surrounding ridges. In addition, the authors discuss gaps in the seismicity distribution along the Central Indian Ridge, that are not closed by their additional data. While all results are clearly described, the interpretation is somewhat short and could easily be enhanced by some more information. Authors Reply: We are thankful for the appreciation of our work and for pointing out the shortcomings. Based on the specific comments provided below we will enhance the discussion section of our manuscript.

Specific comments Referee 1: To expand the discussion of the seismicity, the authors could, for example, include in their interpretative Fig. 11 the age of the oceanic lithosphere to discuss whether age differences and hence differing thermal contraction could be responsible for the intraplate seismicity. Clusters 2 and 3 seem to be located in prolongation of transform zones, so this could support this interpretation.

Authors Reply: We agree that thermal contraction could play a role in the occurrence of these clusters. This was only partially addressed in the discussion (lines 225 and 226) and we intend to discuss this in more detail in the revised version of the manuscript. Referee 1: Fig. 11 could include a colour or symbol coding to distinguish between earthquakes in the global catalogue that were detected and those that were not detected. Authors Reply: The required information was provided in Fig. 8. Yellow and orange symbols are used to discriminate between events from the global catalogue that were detected (and located) by the array and those that were not. Fig. 11 already contains much information related to the interpretation of the results. To keep the maps readable, repeated information were avoided in Fig. 11. However, we decided to combine the information provided in Figs. 8 and 9 in the revised version. Referee 1: A seismic stripe-and-gap pattern in teleseismic and hydroacoustic data have for example been discussed in more detail by Escartin et al. (2008) and by Simao et al.

(2010). They see a relative lack of earthquakes near magmatic centres of spreading segments, whereas segment ends at the Mid-Atlantic Ridge tend to show increased seismicity rates. This observation could provide support for the interpretation of gap 2. A regional seismicity analysis with data just south of the survey area (25 S) is presented by Tsang-Hin-Sun et al. (2016). These authors also find seismic gaps that are even somewhat clearer delineated since they use a hydroacoustic data set with lower detection thresholds. This study could therefore provide additional evidence for the existence of the seismic gaps in the present study area. Authors Reply: We appreciate pointing out the shortcomings of our discussion. The above mentioned references provide important additional information and support for our study and will be included as part of our discussion in the revised manuscript.

Referee 1: Fig. 11 uses 18 years of data from the USGS catalogue. The reviewed ISC bulletin shows many more events in this area for the period 1970-2017. Authors Reply: The additional data from ISC bulletin will be added to the figure. Referee 1: It would be interesting to see whether the seismicity of cluster 4 is swarm-like and occurs in a short time period. Authors Reply: The seismicity of cluster 4 is not swarm-like; the events occur intermittently over a period of 13 months. The details of cluster 4 have been added to Table S2 of the supporting information.

Please also note the supplement to this comment:
https://se.copernicus.org/preprints/se-2020-56/se-2020-56-AC1-supplement.pdf

---

## Author Comment (AC2) · 14 Aug 2020

We would like to thank the reviewers for the time spent on the review and for the helpful comments and constructive suggestions. We discuss below the comments made by referee #2. The comments of the referee are in black and the reply from Authors are provided in blue to facilitate the reading.

General comments Referee 2: The authors use a temporary array on Rodrigues Island to detect and locate (mostly) intraplate earthquakes west of the Central Indian Ridge. This is an entirely new study, which makes the most out of a small array of seismic stations. The analysis is sound; the approach of using beam forming to estimate azimuth

and S-P times to estimate distance is sensible. Authors Reply: We are thankful for the appreciation of our work.

Specific comments Referee 2: It is not clear that these locations provide much new information about seismic gap 2, given the absence of detection of events from the adjacent Egeria FZ. The lack of small events could be attributed instead to poor propagation of Pn and Sn along the path to the array. There is no need for the direct effects of melt on attenuation along the paths; the attenuation is more likely attributable to the presence of thin lithosphere associated with the flow of hotter mantle to the spreading center from the hotspot. Authors Reply: We do not really see this comment in opposition to our conclusions. However, as it is assumed that Pn propagates in the uppermost section of the mantle, a thinned lithosphere, alone, may not have such a significant effect (i.e. attenuation) on the propagation. We, therefore, concluded that partial melting also plays a role. We further agree that this is caused by hotter mantle material between the hotspot and the spreading center. Referee 2: Lines 176-179 appear to be out of place, since they refer to "the event", which is not introduced until lines 186-187. Authors Reply: We thank the referee for pointing out this shortcoming. The paragraph will be shifted after line 192. Referee 2: Also lines 186-191 suggest that array analysis of this example event is shown in figure 7, but that aspect of the event is not illustrated. Authors Reply: A new figure will be provided in the revised version and subsequently the figures will be renumbered in the text as well as the list. Referee 2: Since Figure 9 largely duplicates information shown in Figures 8 and 11, this figure could be used to show regions of uncertainty around each of the locations calculated from uncertainty in azimuth and scatter in S-P picks. Authors Reply: We tried to do this, however, plotting of uncertainties in locations will make the figure unreadable as most of the events occur in clusters. To overcome this, all uncertainties are provided in Table S1 of the supporting information. However, we decided to combine the information provided in Figs. 8 and 9 in the revised version. Referee 2: P picks appear to be pretty clear, but S picks somewhat subjective. Are S picks on each seismogram made independently, or is there iteration between different picks for each

event? Authors Reply: All S picks were made independently and are based on visual inspection of horizontal components as well as vertical where necessary. Referee 2: Cluster 4 is not shown in Figure 10, although it is mentioned in the caption. Authors Reply: Thanks for pointing this out. The figure caption will be modified accordingly.

Please also note the supplement to this comment:
https://se.copernicus.org/preprints/se-2020-56/se-2020-56-AC2-supplement.pdf

---

## Author Response (AR2)

**RESPONSE TO THE INTERACTIVE COMMENT OF REFEREE #1**

We would like to thank the reviewer for the time spent on the review and for the helpful comments and constructive suggestions. The comments of the referee are in bold and the replies from Authors are provided in italics to facilitate the reading.

General comments

**Referee 1: The manuscript presents a thorough array-based analysis of seismicity surrounding Rodrigues Island in the Indian Ocean. Due to its remote location and proximity to the Rodrigues Triple junction, this data set offers interesting insights into intra-plate seismicity of an oceanic plate. The authors exploit the array geometry to estimate locations of 63 events that were not detected by global networks. They suggest a simple, but adequate way of estimating epicentral distance and use backazimuths derived from array analysis. The method is very clearly documented and well-illustrated. The results suggest some well-defined clusters of intra-plate seismicity, that are clearly located away from any of the surrounding ridges. In addition, the authors discuss gaps in the seismicity distribution along the Central Indian Ridge, that are not closed by their additional data. While all results are clearly described, the interpretation is somewhat short and could easily be enhanced by some more information.**

Authors response: *We are thankful for the appreciation of our work and for pointing out the shortcomings. Based on the specific comments provided below we have enhanced the discussion section of our manuscript.*

Authors changes: *The discussion section has been modified, based on the comments from line numbers 257 to 265 and from lines 279 to 283 as shown below.*

Line 257: "*Various mechanisms providing explanation to the cause of intraplate seismicity have been proposed previously. De Long et al. (1977) suggested that different lithospheric ages across the fracture zones, as also observed in the Rodrigues–CIR region, are related to differential subsidence causing stresses and hence earthquakes. Similarly, Collette (1974) and Turcotte (1974) suggested thermal contraction due to different ages of the oceanic crust (Müller et al., 2008) as another*

*possible mechanism. This may be substantiated by the seismicity observed in clusters 2 and 3 as they occur along possible prolongations of fracture zones.*

Line 279: *Based on hydroacoustic data, similar seismic gaps were also observed along the intermediate-spreading Southeast Indian Ridge (Tsang-Hin-Sun et al., 2016) and the slow-spreading Mid-Atlantic Ridge (Escartin et al., 2008; Simao et al. 2010), where events were focused on ridge-segments ends. This uneven distribution of seismicity along CIR, thus, suggests segment-scale variation in lithosphere structure, also observed in Southeast Indian Ridge (Tsang-Hin-Sun et al., 2016) and Mid-Atlantic Ridge (Goslin et al., 2012)."*

Specific comments

**Referee 1: To expand the discussion of the seismicity, the authors could, for example, include in their interpretative Fig. 11 the age of the oceanic lithosphere to discuss whether age differences and hence differing thermal contraction could be responsible for the intraplate seismicity. Clusters 2 and 3 seem to be located in prolongation of transform zones, so this could support this interpretation.**

Authors response: *We agree that thermal contraction could play a role in the occurrence of these clusters. This was only partially addressed in the discussion (lines 225 and 226) and has now been discussed in more detail in the revised version of the manuscript.*

Authors changes: *Pease refer to the below text. Thermal contraction has been discussed between lines 259 and 262 and has been modified to:*

"*Similarly, Collette (1974) and Turcotte (1974) suggested thermal contraction due to different ages of the oceanic crust (Müller et al., 2008) as another possible mechanism. This may be substantiated by the seismicity observed in clusters 2 and 3 as they occur along possible prolongations of fracture zones.*"

**Referee 1: Fig. 11 could include a colour or symbol coding to distinguish between earthquakes in the global catalogue that were detected and those that were not detected.**

Authors response: *The required information was provided in Fig. 8. Yellow and orange symbols are used to discriminate between events from the global catalogue that were detected (and located) by the array and those that were not. Fig. 11 already contains much information related to the interpretation of the results. To keep the maps readable, repeated information were avoided in Fig. 11. However, we have combined the information provided in Figs. 8 and 9 in the revised version into new Fig. 9. The old Fig. 8 has been provided in supporting information as Fig. S3. (We also provide a new Fig. 8, as described in response to reviewer 2.)*

Authors changes: *Please see new Fig. 9 in the revised manuscript and new Fig. S3 in supporting information.*

**Referee 1: A seismic stripe-and-gap pattern in teleseismic and hydroacoustic data have for example been discussed in more detail by Escartin et al. (2008) and by Simao et al. (2010). They see a relative**

**lack of earthquakes near magmatic centres of spreading segments, whereas segment ends at the Mid-Atlantic Ridge tend to show increased seismicity rates. This observation could provide support for the interpretation of gap 2. A regional seismicity analysis with data just south of the survey area (25 S) is presented by Tsang-Hin-Sun et al. (2016). These authors also find seismic gaps that are even somewhat clearer delineated since they use a hydroacoustic data set with lower detection thresholds. This study could therefore provide additional evidence for the existence of the seismic gaps in the present study area.**

Authors response: *We appreciate pointing out the shortcomings of our discussion. The above mentioned references provide important additional information and support for our study and have been included as part of our discussion in the revised manuscript.*

Authors changes: *References discussed in the text and added in the list*

*Escartín, J., Smith, D. K., Cann, J., Schouten, H., Langmuir, C. H., and Escrig, S.: Central role of detachment faults in accretion of slow-spreading oceanic lithosphere, Nature, 455(7214), 790–794, 2008.*

*Goslin, J., Perrot, J., Royer, J. Y., Martin, C., Lourenço, N., Luis, J., Dziak, R. P., Matsumoto, H., Haxel, J., Fowler, M. J., and Fox, C. G.: Spatiotemporal distribution of the seismicity along the Mid-Atlantic Ridge north of the Azores from hydroacoustic data: Insights into seismogenic processes in a ridge–hot spot context, Geochem. Geophys. Geosyst., 13(2), 2012.*

*Müller, R. D., Sdrolias, M., Gaina, C., and Roest, W. R.: Age, spreading rates, and spreading asymmetry of the world's ocean crust, Geochem., Geophys., Geosyst., 9(4), 2008.*

*Simao, N., Escartin, J., Goslin, J., Haxel, J., Cannat, M., and Dziak, R.: Regional seismicity of the Mid-Atlantic Ridge: observations from autonomous hydrophone arrays, Geophys. J. Int., 183(3), 1559–1578, 2010.*

*Tsang-Hin-Sun, E., Royer, J. Y., and Perrot, J.: Seismicity and active accretion processes at the ultraslow-spreading Southwest and intermediate-spreading Southeast Indian ridges from hydroacoustic data, Geophys. J. Internat., 206(2), 1232–1245, 2016.*

*Leva, C., Rümpker, G., Link, F. and Wölbern, I.: Mantle earthquakes beneath Fogo volcano, Cape Verde: Evidence for subcrustal fracturing induced by magmatic injection. J. Volcanol. Geotherm. Res., 386, 106672, 2019.*

**Referee 1: Fig. 11 uses 18 years of data from the USGS catalogue. The reviewed ISC bulletin shows many more events in this area for the period 1970-2017.**

Authors response: *The additional data from ISC bulletin (magenta symbols in the Figure below) have not been included in our study because the location accuracy for the ISC events is significantly lower than for the events in the USGS catalogue (as judged from the spread of the events shown in the Figure below). This is the reason for focusing on the events provided by the USGS catalogue.*

[Figure]

Authors changes: *None.*

**Referee 1: It would be interesting to see whether the seismicity of cluster 4 is swarm-like and occurs in a short time period.**

Authors response: *The seismicity of cluster 4 is not swarm-like; the events occur intermittently over a period of 13 months.*

Authors changes: *The details of cluster 4 have been added to Table S2 of the supporting information.*

**RESPONSE TO THE INTERACTIVE COMMENT OF REFEREE #2**

*We would like to thank the reviewer for the time spent on the review and for the helpful comments and constructive suggestions. The comments of the referee are in bold and the reply from Authors are provided in italics to facilitate the reading.*

General comments

**Referee 2: The authors use a temporary array on Rodrigues Island to detect and locate (mostly) intraplate earthquakes west of the Central Indian Ridge. This is an entirely new study, which makes the most out of a small array of seismic stations. The analysis is sound; the approach of using beam forming to estimate azimuth and S-P times to estimate distance is sensible.**

Authors response: *We are thankful for the appreciation of our work.*

Specific comments

**Referee 2: It is not clear that these locations provide much new information about seismic gap 2, given the absence of detection of events from the adjacent Egeria FZ. The lack of small events could be attributed instead to poor propagation of Pn and Sn along the path to the array. There is no need for the direct effects of melt on attenuation along the paths; the attenuation is more likely attributable to the presence of thin lithosphere associated with the flow of hotter mantle to the spreading center from the hotspot.**

Authors response: *We do not really see this comment in opposition to our conclusions. However, as it is assumed that Pn propagates in the uppermost section of the mantle, a thinned lithosphere, alone, may not have such a significant effect (i.e. attenuation) on the propagation. We, therefore, concluded that partial melting also plays a role. We further agree that this is caused by hotter mantle material between the hotspot and the spreading center.*
Authors changes: *None.*

**Referee 2: Lines 176-179 appear to be out of place, since they refer to "the event", which is not introduced until lines 186-187.**

Authors response: *We thank the referee for pointing out this shortcoming.*
Authors changes: *The paragraph has been shifted to line 216.*

**Referee 2: Also lines 186-191 suggest that array analysis of this example event is shown in figure 7, but that aspect of the event is not illustrated.**

Authors response: *We thank the referee for pointing out this shortcoming.*

Authors changes: *A new Fig. 8 has been provided in the revised version to show the analysis of the event described and subsequently the figures have been renumbered in the text as well as the list.*

**Referee 2: Since Figure 9 largely duplicates information shown in Figures 8 and 11, this figure could be used to show regions of uncertainty around each of the locations calculated from uncertainty in azimuth and scatter in S-P picks.**

Authors response: *We tried to do this, however, plotting of uncertainties in locations will make the figure unreadable as most of the events occur in clusters. To overcome this, all uncertainties are provided in Table S1 of the supporting information.*

Authors changes: *We have combined the information provided in Figs. 8 and 9 in the revised version into new Fig. 9. The old Fig. 8 has been provided in supporting information as Fig. S3.*

**Referee 2: P picks appear to be pretty clear, but S picks somewhat subjective. Are S picks on each seismogram made independently, or is there iteration between different picks for each event?**

Authors response: *All S picks were made independently and are based on visual inspection of horizontal components as well as vertical where necessary.*

Authors changes: *New text in line 138-139 has been added "All S picks were made independently and are based on visual inspection of horizontal components as well as vertical where necessary."*

**Referee 2: Cluster 4 is not shown in Figure 10, although it is mentioned in the caption.**

Authors response: *We thank the referee for pointing out this shortcoming.*

Authors changes: *The figure caption has been modified to "Variation in longitude (a) and latitude (b) of the events detected and located using Rodrigues array. Monthly distribution of the events is shown in (c). The red solid line in (b) marks the events for Cluster 1 (~120 km north-east of Rodrigues Island), Cluster 2 (~140 km north-west of Rodrigues Island), and Cluster 3 (~265 km south of Rodrigues Island)."*

**ADDITIONAL MODIFICATIONS**

Authors response: *In addition to the requests by the reviewers we also modified the magnitude calculations. In the original submitted version of the manuscript, local magnitudes were based on amplitudes obtained at or near the S-wave arrival time. In the revised version we adhere to the more conventional approach and determine the local magnitude from the maximum (absolute) amplitude of the displacement seismogram on the two horizontal components (see paragraph 2.3). This approach leads to slightly larger magnitude values in comparison to those reported in the original submitted manuscript. The differences in magnitudes for selected events (Tables 1 and 2) is also reduced. Remaining discrepancies, however, may be best explained by differences in the radiation directions and/or attenuation effects, as suggested in the discussion.*

Authors changes: *Text between lines 95 and 99 has been modified to 'In order to calculate the magnitude of an event, we account for the sensitivity of the recording system (CUBE datalogger and 1 Hz MARK sensor) and integrate the velocity seismogram, which is then convolved with the Wood–Anderson transfer function to obtain the ground displacement in nanometers. Magnitudes are determined based on the maximum (absolute) amplitude of the horizontal components using all available stations for which the recordings show a clear (dominant) S-phase. Again, the mean and the SD of the magnitude value is calculated. As most of the events are located well within 1000 km radius, we use the relation given by Havskov and Ottemoller (1999) in SEISAN package to determine the local magnitude.'*

*Additional explanation has been provided between line 205 and 207 'Amplitude variations related to the different radiation directions relevant for regional (at Rodrigues island) and teleseismic recordings may also play a role. In addition, the amplitudes for the dominantly horizontal Pn raypaths from the recordings at Rodrigues may be more affected by regional attenuation processes, as described further in the next section.'*

*The magnitude values have been modified in Tables 1, S1 and S2 and the relevant figures are modified accordingly.*

[revised manuscript text omitted]